# OpenReview forum: "SafeWatch: An Efficient Safety-Policy Following Video Guardrail Model with Transparent Explanations"
_ICLR.cc/2025/Conference — ICLR 2025 Poster_

### Official Review · Reviewer_PM58 · 2024-10-28

**Soundness:** 3
**Presentation:** 3
**Contribution:** 3
**Rating:** 6
**Confidence:** 3

**Summary:**

The paper proposed a safety aware video understanding benchmark, including 2M human verified videos. The unsafe scenarios are separated into 6 classes. The authors also design a pipeline for automatically data generation. For the video understanding model, the authors propose the Parallel Equivalent Policy Encoding and Policy-Aware Adaptive Pruning to encode the Safety Policy Guidelines and reduce the redundancy. The result of the trained model is good comparing to both close- and open- source models.

**Strengths:**

1. The proposed dataset is novel and the data curation procedure is well-organized.
2. The proposed Parallel Equivalent Policy Encoding and Policy-Aware Adaptive Pruning can effectively encode the Safety Policy Guidelines and reduce the redundancy of video tokens.
3. The results are good.

**Weaknesses:**

1. Dataset Construction:
       (1) The release and separation of the dataset is a concern. The author can only provide links to publicly available sources and annotations. But it is common that the link may fail.I understand that this is something unavoidable, but it will undoubtedly reduce the frequency of use and impact of this dataset.
       (2) 2M videos for human verification is a huge effort, the authors don't provide any details of the procedure.

2. Model training：
      (1) Some of the training procedure is ambiguous, there should be more details about Preference Post-tuning procedure.

**Questions:**

1. The author mentioned that Human Verification is used for curating the 2M video benchmark. Can the authors make a detailed description of the Human Verification procedure?

2. More procedure should be enclosed in the appendix.

---

> ### Author Response · Authors · 2024-11-22
> **Response to Reviewer PM58 (Part 1)**
>
> Dear Reviewer PM58,
>
> Thank you so much for your valuable suggestions to help improve our paper! We deeply appreciate your recognition of the novelty and effectiveness of our work! Below, we have provided a detailed, point-by-point response to your questions and hope these can help address your concerns.
>
>
> > **W1**: Dataset Construction: (1) The release and separation of the dataset is a concern. The author can only provide links to publicly available sources and annotations. But it is common that the link may fail.I understand that this is something unavoidable, but it will undoubtedly reduce the frequency of use and impact of this dataset. (2) 2M videos for human verification is a huge effort, the authors don't provide any details of the procedure.
>
>
> We sincerely appreciate the reviewer’s concerns regarding the public impact of our dataset Firstly, We acknowledge that links to publicly available sources may occasionally fail, potentially reducing the dataset's usability over time. However, this approach is necessary given the large scale and safety-related nature of this dataset. It aligns with the practices of several widely adopted datasets in the CV and NLP communities such as LAION [1] and Kinetics [2], which also provide links to publicly available sources instead of directly hosting the data. This approach is particularly common in video datasets, where the storage requirements are significantly larger. For example, Kinetics provides only URLs and invites practitioners to download videos using provided scripts, a practice we have followed for similar reasons. Furthermore, while our proposed guardrail model SafeWatch has been trained on a dataset containing sexually explicit, harmful, and violent content, we will not release SafeWatch-Bench in its entirety. Instead, we will release a carefully curated subset that meets the following criteria: (1) Extremely unsafe content, such as instances of child abuse, has been removed; (2) All human identities have been blurred to protect privacy; (3) Only content permitted under copyright laws is included, ensuring the data cannot be misused. For the releasable subsets, we are considering maintaining a local copy of certain data and providing direct access to the videos in a manner that complies with legal regulations. This will ensure the reproducibility of our dataset while preserving the integrity and impact of our work. We will ensure to handle this carefully and detail it in our dataset impact statement following your suggestions.
>
> Regarding the second question, we totally agree with the reviewer that 2M videos for human verification are a huge effort, and in practice we do not ask human verifiers to review all 2M videos, instead, we only ask them to review one video in each batch of the 200K videos that we used for training the model, as well as all the 1420 videos in the test set that we used for evaluation, keeping the manual verification effort manageable.  For the remaining videos in the 2M dataset, annotations are generated exclusively using the multi-agent pipeline without human verification. While this may introduce a certain error rate in the remaining dataset, we have demonstrated the effectiveness of our closed-loop pipeline in safety-related video annotation below that can effectively ensure the overall dataset's quality.
>
> We provide detailed statistics of our dataset in Table-r. 1 below and more details regarding the data curation process in Table-r. 2. Specifically, we first group the 200K videos based on similar sources and types (e.g. same user ID or benchmark subcategory) in a batch (we use a batchsize of 64); (2) Then we run the multi-agent discussion pipelines event-by-event to annotate each video in the batch (all prompts provided in Appendix.B8); (3) Then we ask human verifiers to sample one video from each batch to review their explanation quality and decide whether to reject the batch and re-annotate. Specifically, grouping similar videos in a batch ensures they have similar annotation quality or shared issues, improving efficiency and reducing manual costs. If a batch has been rejected twice, then we discard this batch to exclude it from the dataset. As shown in Table-r. 2, human verifiers validate 3247 batches in total, with a low first-time rejection rate of 13.79%, demonstrating the effectiveness of our pipeline. Among the re-annotated batches, 23.7% (6784 videos) were rejected again and discarded to ensure the overall quality of the dataset.
>
> [1] Schuhmann, C., Beaumont, R., Vencu, R., Gordon, C., Wightman, R., Cherti, M., ... & Jitsev, J. (2022). Laion-5b: An open large-scale dataset for training next generation image-text models. Advances in Neural Information Processing Systems, 35, 25278-25294.
>
> [2] Kay, W., Carreira, J., Simonyan, K., Zhang, B., Hillier, C., Vijayanarasimhan, S., ... & Zisserman, A. (2017). The kinetics human action video dataset. arXiv preprint arXiv:1705.06950.

---

> > ### Author Response · Authors · 2024-11-22
> > **Response to Reviewer PM58 (Part 2)**
> >
> > Table-r. 1 Additional statistics of the SafeWatch-Bench Dataset. We provide details for both training and testing configurations, including total videos, average video length (seconds), explanation length (word count), number of events, and the ratio of multi-labeled videos.
> > | Datset Configuration                  | **Training**                              | **Testing**                               |
> > |-----------------------------|-------------------------------------------|-------------------------------------------|
> > | Total Videos            | 199604       | 1420        |
> > | Average Video Length    | 57.49 | 61.12  |
> > | Average Explanation Length | 80.20       | 73.16         |
> > | Average Number of Events      | 7.33           | 7.40           |
> > | Average Ratio of Multi-label        | 24.70          | 28.57           |
> >
> > Table-r. 2 Statistics of the multi-agent data curation process
> > |                   | #Batches                          | Avg #Iterations                          |
> > |-----------------------------|-------------------|------------------------|
> > | Total            | 3247      | 1.89      |
> > | Rejected    | 448 | 2.30      |
> > | Discarded | 106       | 2.87      |
> >
> >
> > > **W2**: Model training： (1) Some of the training procedure is ambiguous, there should be more details about Preference Post-tuning procedure.
> >
> >
> > We totally agree with the reviewer that more details regarding model training are critical for understanding the effectiveness of our work. In response, we have provided a point-by-point explanation of our training procedure below.
> >
> > + **Dataset partition:** For more effective training and evaluation, we use the 200K videos verified by humans via batch sampling and select 1420 videos to consist the testing set for benchmarking (830 real-world videos, 590 generative videos), and use the rest of 199604 videos for training. Specifically, we aim to ensure the diversity of the videos and balanced coverage of all categories in the test set (also shown in Fig. 10 in the appendix; some case studies in Fig. 16, 17, 20).
> > + **Data usage in each training stage:** We detail the data usage for each of the three fine-tuning stage, and we demonstrate the corresponding model's performance in each stage in Table-r. 3:
> >     + **Multi-task Guardrail Training:** In this stage, we randomly sample 80K guardrail-related videos we collected and 30K normal videos in ShareGPT4Video annotated by GPT-4v, and then we augment the original annotations into multiple tasks including captioning, VQA, and video guardrails, resulting in 220K training samples. This stage aims to train the model to develop general guardrail capabilities while preserving a broad understanding of general video content, effectively mitigating catastrophic forgetting and overfitting to guardrail-specific videos.
> >     + **Adaptive-Pruning Training:** In this stage, we solely fine-tune the model on all the 199K guardrail-related videos using four types of guardrail task prompts specified in Appendix.B8. This stage aims to train the model to extract essential information from a subset of more informative video tokens via PAP and downstream the model for specialized guardrail tasks.
> >     + **Preference Post-tuning:** In this stage, we aim to further improve the quality of explanations. Specifically, we curate the rejected explanations from two sources (1) **offline collection**: the non-specific or overly long explanations that we discarded during the multi-agent propose-discuss pipeline; (2) **online sampling:** we run the model from the previous stage to infer through 5K diverse videos in the training set and collect those samples with wrong answer. And we use the corresponding ground-truth explanations as the chosen pair. This process results in 60K problem-centric preference pairs and we fine-tuned the model using DPO.
> >
> > + **Additional Dataset Statistics:** We provide more statistics regarding our dataset and training procedure. Specifically,
> >      + The **average video length** in the training and testing set is 57.49 and 61.12 secs, respectively (more detailed distribution in Fig. 11 in the appendix), with the longest video up to 90 minutes, ensuring comprehensive coverage of unsafe videos with different scales (e.g. short videos, movies).
> >      + The **average explanation length** in the training and testing set is 80.2 and 73.16 words, ensuring a detailed and in-depth reasoning of the guardrail result (more detailed distribution in Fig. 12, two case studies in Fig. 16, 17 in the appendix).
> >      + The **average number of events** generated by our safe-aware event sampler model in the training and testing set is 7.33 and 7.4. This reflects the high complexity and challenging nature of the videos in the guardrail task.
> >      + The **average ratio of multi-labeled videos** (i.e., videos flagged with multiple guardrail categories) is 24.7% in the training set and 28.57% in the testing set, further demonstrating the dataset's diversity and challenging nature.

---

> > > ### Author Response · Authors · 2024-11-22
> > > **Response to Reviewer PM58 (Part 3)**
> > >
> > > + **Training strategy:** We fully fine-tune all the layers of the model for all three fine-tuning stages including the preference post-tuning stage with 8 H100 GPUs. While fully fine-tuning is critical in the first two stages, for stage three, we observed that LoRA fine-tuning with rank 128 achieves similar guardrail accuracy compared to full fine-tuning. However, the explanation quality is slightly higher when using fully fine-tuning, which is why we opted for this approach for all three stages.
> > >
> > > Table-r. 3 Performance of SafeWatch during each training stage
> > > | Model              | Accuracy | Explanation | Adaptability |
> > > |--------------------|--------------------------|---------------------------|--------------------------|
> > > | InternVL2-8B       | 29.1                   | 5.25                     | 26.8                    |
> > > | Stage-1      | 63.9              | 5.84               | 69.7                |
> > > | Stage-2      | 67.3              | 6.12                | 74.9                |
> > > | Stage-3      | **72.6**              | **7.17**                | **78.0**                |
> > >
> > >
> > > > **Q1**: The author mentioned that Human Verification is used for curating the 2M video benchmark. Can the authors make a detailed description of the Human Verification procedure?
> > >
> > >
> > > We thank the reviewer for this thoughtful question! We have detailed the human verification procedure in our response to W1. Specifically, instead of verifying all 2M videos, we asked human verifiers to examine the 200K videos that we used for training and testing. For the training set, we group similar videos together and annotate them in batches where human verifiers verify them via _batch sampling_, which allows us to efficiently validate the annotation quality for each batch. This approach is both statistically reliable and cost-effective, ensuring high-quality annotations without requiring an exhaustive review of every video. As for the test set, we ask human verifiers to carefully review every video and make corrections when necessary, as absolute accuracy is critical for meaningful evaluation and comparison across models. This approach ensures the quality of the data used for training and evaluation while keeping the manual labeling costs manageable. For more details, please refer to our response to W1 above.
> > >
> > >
> > >
> > > > **Q2**: More procedure should be enclosed in the appendix.
> > >
> > >
> > > Thank you for this thoughtful suggestion! In response, we have expanded the appendix following your suggestion to include additional details about our model and dataset, fine-grained results, dataset statistics, training pipeline, case studies, and all prompts used for data curation and evaluation.
> > >
> > > Specifically:
> > >
> > > - **Appendix A: Ablation Studies and Detailed Results**
> > >   1. Investigation of the individual contributions of each novel component and training stage, presented in **Tables 6-9**.
> > >   2. In-depth analysis of the underlying principles of **PEPE** and **PAP**, detailed in **Table 10** and **Figure 6-7**.
> > >   3. Fine-grained results for all experiments, including:
> > >      - Model performance across different subcategories (**Figure 8**).
> > >      - Results for four types of prompting tasks (**Tables 11–14**).
> > >      - Results on existing datasets (**Table 11**) and new policy categories (**Table 12**).
> > >      - Comparison of explanation quality across training stages (**Figure 9**).
> > >
> > > - **Appendix B: Dataset Overview and Case Studies**
> > >   1. Comprehensive dataset statistics (**Table 20**).
> > >   2. Detailed dataset distribution (**Figures 10–14**).
> > >   3. A case study of the multi-agent data annotation pipeline (**Figure 15**).
> > >   4. Two case studies of dataset annotations (**Figures 16 and 17**).
> > >   5. Case studies on the model's guardrail performance (**Figures 18 and 19**).
> > >   6. A case study of generative video samples in SafeWatch-Bench (**Figure 20**).
> > >   7. All prompts used for data curation and evaluation (**Appendix B8**).
> > >
> > > - **Appendix C: Additional Related Works**
> > >   We provide a more comprehensive discussion of related works in this section.
> > >
> > > These additions offer a detailed and transparent account of our methods, ensuring that our work is fully reproducible and deployable. We hope these expanded details address your concerns and facilitate a deeper understanding of our approach.
> > >
> > > We hope the additional details about our dataset and training pipeline, along with the expanded explanations, have addressed your concerns. Please don’t hesitate to let us know if there’s anything else you’d like to discuss or if further clarification is needed.
> > >
> > > If all your concerns have been resolved, we would sincerely appreciate it if you could consider raising your evaluation score to help us share this work with the broader community.
> > >
> > > Thank you again for your time and thoughtful feedback!
> > >
> > > Best regards,
> > >
> > > Submission #9949 Authors

---

> > > > ### Author Response · Authors · 2024-12-02
> > > >
> > > > Dear Reviewer PM58,
> > > >
> > > > Thank you again for kindly acknowledging the novelty and contribution of our proposed algorithm and dataset! We deeply appreciate your valuable suggestions and thoughtful feedback, which have been instrumental in helping us improve our work. As Dec 2nd is the last day that reviewers can post a message, we wanted to kindly follow up to ensure that our responses have adequately addressed your concerns. We have carefully considered the points you raised and have provided detailed explanations and additional experimental results in our responses above.
> > > >
> > > > We sincerely hope our responses have addressed your concerns, and we are more than happy to address any further questions you may have. Thank you once again for your valuable time and effort in helping us improve our work!

---

### Official Review · Reviewer_YtKs · 2024-10-31

**Soundness:** 3
**Presentation:** 4
**Contribution:** 3
**Rating:** 8
**Confidence:** 3

**Summary:**

The authors of the paper propose a multimodal large language model (MLLM) called SafeWatch, designed to follow customized safety policies and provide multi-label video guardrail categorical outputs with answer explanations in a zero-shot manner. They also introduce SafeWatch-Bench, a large-scale video guardrail benchmark containing over 2 million videos spanning 6 safety broad categories and covering over 30 finer-grained risk categories to ensure comprehensive coverage of potential safety scenarios.

The technical contributions include:
- Model Design: The authors introduce two key plug-and-play modules: Parallel Equivalent Policy Encoding (PEPE) and Policy-Aware Adaptive Pruning (PAP).
  - PEPE mitigates high latency from extensive input contexts and policy positional bias by dividing lengthy safety guidelines into independent chunks encoded in parallel with equal importance.
  - PAP, on the other hand, reduces latency by selecting the most relevant visual tokens for each policy while discarding those with low relevance.

- Data: Each instance in SafeWatch-Bench is annotated with multi-label guardrail categories and detailed explanations. The dataset includes 2 million videos—both real-world and generative from various SOTA models—comprising an instruction-tuning set and a test set of 1K hand-selected, high-quality annotated videos across subcategories.

- Training strategy: The authors fine-tune InternVL2-8B with their modeling changes on this new data via three stages, i.e., multi-task training, adaptive-pruning training, and preference post-tuning.
  - Stage 1: Only PEPE is trained during this stage on a large corpus of unsafe videos, as well as traditional VQA and captioning tasks on normal videos.
  - Stage 2: Both PEPE and PAP are fine-tuned on guardrail tasks.
  - Stage 3: Preference pairs are curated to enable the preference post-tuning.

**Strengths:**

1. Their model, SafeWatch, outperforms SOTA video guardrails on SafeWatch-Bench by 19.6% and on existing benchmarks by 15.4%, while reducing inference costs by an average of 25%. SafeWatch also demonstrates strong policy-following abilities, outperforming baselines by 20% in zero-shot adaptability to new policies. Additionally, both LLM-as-a-judge and human evaluators confirm the high quality of the explanations provided by SafeWatch.
2. The design choices are well-founded, following best practices for efficient MLLM construction.
3. This is an important area of study, with meaningful contributions (if these contributions are reproducible).

**Weaknesses:**

1. Missing Evaluation: The evaluation of the Safety-aware Event Sampling step is absent. This step should be crucial for model performance, as the authors used TransnetV2 to segment videos into safety-aware events, sampling a single frame per event for further MLLM processing.
2. Dataset Clarity: The dataset’s specifics and its exact use in model training remain unclear. For instance, there is no information on the average video length or the typical length of an explanation in SafeWatch-Bench. Additionally, the quality of the SafeWatch-Bench test set is not fully addressed, which is particularly important for an evaluation dataset.
3. Reproducibility Concerns: Reproducibility in data collection and model training is questionable. For instance, Section 4.2 on "multi-agent consensus video annotation" provides a basic idea of the processes but lacks sufficient detail for replication (e.g., missing the prompts used, configurations such as the number of frames used for each model, etc.). Additional issues are noted in the "Questions" section.

**Questions:**

1. Could you provide some evaluation results for Safety-aware Event Sampling? Currently, its effectiveness or limitations are unclear.
2. Is SafeWatch-Bench truly a video benchmark? Specifically, does it truly require reasoning across multiple frames for a model to achieve high performance?
3. It appears that humans do not directly provide annotations for SafeWatch-Bench; instead, annotations are model-generated, with human reviewers checking if re-annotation is necessary. In the caption for Figure 2, you mention that the 1K test set has high-quality annotations. Were these test set annotations created directly by humans, or were they produced via the multi-agent propose-discuss-consensus pipeline?
4. Could you describe how the preference pairs were curated for the preference post-tuning stage? Additionally, how were the challenging benign examples—those easily identified by humans but likely to mislead guardrail models—selected? A detailed explanation of these curation processes would be helpful.
5. What is the quality of the synthetic videos generated by the GenAI models? Are they accurately aligned with the unsafe prompts? Was any video filtering applied to filter out misaligned videos?
6. Could you clarify your model training data recipe? Are the unsafe videos used in stage-1 training and the guardrail tasks in stage-2 training drawn from the instruction-tuning dataset within SafeWatch-Bench? Additionally, how many samples are used for each stage, task, and dataset?
7. Which layers are tuned in the preference post-tuning stage?
8. In Table 3 of Appendix A.1, there is a column named "Temporal location". What does it mean?
9. What is the SFT Baseline mentioned in Figure 5 and Table 5?
10. It was claimed that prior datasets lack detailed descriptions of the videos, suggesting that SafeWatch-Bench offers a detailed description for each video. Is that correct?
11. There is unsafe content in SafeWatch-Bench. Will SafeWatch and SafeWatch-Bench be released? If so, how do you plan to ensure their proper use?

Minor comments:
Line 749 has placeholder text.

**Details Of Ethics Concerns:**

The authors claim data contributions, but there is unsafe content in SafeWatch-Bench.

---

> ### Author Response · Authors · 2024-11-22
> **Response to Reviewer YtKs (Part 1)**
>
> Dear Reviewer YtKs,
>
> Thank you so much for your valuable suggestions to help improve our paper! We deeply appreciate your recognition of the novelty of our work and its technical contribution to addressing the risks brought by the rapid proliferation of MLLMs. Below, we have provided a detailed, point-by-point response to your questions and hope these could help address your concerns.
>
>
> > **W1**: Missing Evaluation: The evaluation of the Safety-aware Event Sampling step is absent. This step should be crucial for model performance, as the authors used TransnetV2 to segment videos into safety-aware events, sampling a single frame per event for further MLLM processing.
>
> Thank you for this valuable suggestion! We totally agree with the reviewer that the safety-aware event sampling model is a crucial component for the guardrail performance, and we have provided an additional evaluation of it below in Table-r. 1. Specifically, to reduce the heavy annotation workload, we first observe the connection between safety event segmentation and shot boundary detection, where we find that while being similar, our task differs in that multiple consecutive shots can belong to the same event. Noting this difference, we first adopt a SOTA shot boundary detection model AutoShot [1] to perform an initial segmentation on 742 videos sampled from the SafeWatch-Bench training set. These videos were carefully selected to ensure a comprehensive representation of all the unsafe video categories. Next, we ask human verifiers to review the segmented results and make corrections when necessary (primarily merging segments). This approach allowed us to produce sufficient high-quality frame annotations tailored for the safety event sampling task. Then we further split 74 videos as a test set and followed AutoShot to train our model based on TransnetV2. The evaluation results are presented in Table-r.1. Specifically, our model outperforms other models on the safety-aware event sampling task in terms of F1 score. Notably, our model achieves much higher precision compared to other general shot boundary detection models, reflecting its suitability for this specific safety guardrail task.
>
>
> Table-r. 1 Evaluation of Safety-aware Event Sampling Model
> |                   | F1 scores                          |
> |-----------------------------|--------------|
> | TransnetV2           | 82.4 |
> | AutoShot   | 87.8 |
> | Safety-aware Event Sampling   | 94.6 |
>
> We hope this additional result and explanation have addressed your concerns, and we have updated the revision following your suggestions.
>
> [1] Zhu, W., Huang, Y., Xie, X., Liu, W., Deng, J., Zhang, D., ... & Liu, J. (2023). Autoshot: A short video dataset and state-of-the-art shot boundary detection. In Proceedings of the IEEE/CVF Conference on Computer Vision and Pattern Recognition (pp. 2238-2247).

---

> > ### Author Response · Authors · 2024-11-22
> > **Response to Reviewer YtKs (Part 2)**
> >
> > > **W2**: Dataset Clarity: The dataset’s specifics and its exact use in model training remain unclear. For instance, there is no information on the average video length or the typical length of an explanation in SafeWatch-Bench. Additionally, the quality of the SafeWatch-Bench test set is not fully addressed, which is particularly important for an evaluation dataset.
> >
> > We totally agree with the reviewer that more details on the datasets specifics and model training are critical for understanding the mechanism and effectiveness of our work. In response, we have provided additional details on the dataset statistics and training procedure in Table-r. 2 below, as well as Fig. 10-20 in the updated appendix.
> >
> > + **Dataset partition:** For more effective training and evaluation, we use the 200K videos verified by humans and select 1420 videos to consist the testing set for benchmarking (830 real-world videos, 590 generative videos), and use the rest of 199604 videos for training. Specifically, we aim to ensure the diversity of the videos and a balanced coverage of all categories in the test set (a more detailed sample distribution is shown in Fig. 10, data sample case study in Fig. 16-20 in the appendix).
> > + **Data usage in each training stage:** We detail the data usage for each fine-tuning stages and demonstrate the corresponding model's performance in each stage in Table-r. 3:
> >     + **Multi-task Guardrail Training:** In this stage, we randomly sample 80K guardrail-related videos we collected and 30K normal videos in ShareGPT4Video annotated by GPT-4v, and then we augment the original annotations into multiple tasks including captioning, VQA, and video guardrails, resulting in 220K training samples. This stage aims to train the model to develop general guardrail capabilities while preserving a broad understanding of general video content, effectively mitigating catastrophic forgetting and overfitting to guardrail-specific videos.
> >     + **Adaptive-Pruning Training:** In this stage, we solely fine-tune the model on all the 199K guardrail-related videos using four types of guardrail task prompts specified in Appendix.B8. This stage aims to train the model to extract essential information from a subset of more informative video tokens via PAP and downstream the model for specialized guardrail tasks.
> >     + **Preference Post-tuning:** In this stage, we aim to further improve the quality of explanations. Specifically, we curate the rejected explanations from two sources (1) **offline collection**: the non-specific or overly long explanations that we discarded during the multi-agent propose-discuss pipeline; (2) **online sampling:** we run the model from the previous stage to infer through 5K diverse videos in the training set and collect those samples with wrong answer. And we use the corresponding ground-truth explanations as the chosen pair. This process results in 60K problem-centric preference pairs and we fine-tuned the model using DPO.
> >
> > + **Additional Dataset Statistics:** We provide more statistics to better understand our dataset and training procedure. Specifically,
> >      + The **average video length** in the training and testing set is 57.49 and 61.12 secs, respectively (more detailed distribution in Fig. 11 in the appendix), with the longest video up to 90 minutes, ensuring comprehensive coverage of unsafe videos with different scales (e.g. short videos, movies).
> >      + The **average explanation length** in the training and testing set is 80.2 and 73.16 words, ensuring detailed and in-depth reasoning of the guardrail result (more detailed distribution in Fig. 12, two case studies in Fig. 16, 17 in the appendix).
> >      + The **average number of events** generated by our safe-aware event sampler model in the training and testing set is 7.33 and 7.4. This reflects the high complexity and challenging nature of the videos in the guardrail task.
> >      + The **average ratio of multi-labeled videos** (i.e., videos flagged with multiple guardrail categories) is 24.7% in the training set and 28.57% in the testing set, further demonstrating the dataset's diversity and challenging nature.
> >
> > + **Quality of test set:** We first provide a detailed description of the data annotation and human verification procedure for the training set in our response to W3 below. Specifically, we ask human verifiers to validate the quality of 3247 videos annotated by the multi-agent pipeline, where they have accepted 86\% batches annotated by the pipeline to achieve human-like quality, and by re-annotating the rejected batches, we have finally achieved a 96.73\% acceptance rate. As for the 1420 videos in the test set, which are required to be absolutely accurate, we ask human verifiers to review each video and make corrections when necessary. This meticulous process ensures high-quality annotations for the test set. To further illustrate the quality of our annotations, we have included several case studies in Figures 15-17 in the appendix.

---

> ### Author Response · Authors · 2024-11-22
> **Response to Reviewer YtKs (Part 3)**
>
> Table-r. 2 Additional statistics of the SafeWatch-Bench Dataset. We provide details for both training and testing configurations, including total videos, average video length (seconds), explanation length (word count), number of events, and the ratio of multi-labeled videos.
> | Datset Configuration                  | **Training**                              | **Testing**                               |
> |-----------------------------|-------------------------------------------|-------------------------------------------|
> | Total Videos            | 199604       | 1420        |
> | Average Video Length    | 57.49 | 61.12  |
> | Average Explanation Length | 80.20       | 73.16         |
> | Average Number of Events      | 7.33           | 7.40           |
> | Average Ratio of Multi-label        | 24.70          | 28.57           |
>
>
> Table-r. 3 Performance of SafeWatch during each training stage
> | Model              | Accuracy | Explanation | Adaptability |
> |--------------------|--------------------------|---------------------------|--------------------------|
> | InternVL2-8B       | 29.1                   | 5.25                     | 26.8                    |
> | Stage-1      | 63.9              | 5.84               | 69.7                |
> | Stage-2      | 67.3              | 6.12                | 74.9                |
> | Stage-3      | **72.6**              | **7.17**                | **78.0**                |
>
>
>
> > **W3**: Reproducibility Concerns: Reproducibility in data collection and model training is questionable. For instance, Section 4.2 on "multi-agent consensus video annotation" provides a basic idea of the processes but lacks sufficient detail for replication (e.g., missing the prompts used, configurations such as the number of frames used for each model, etc.). Additional issues are noted in the "Questions" section.
>
> Thank you for this valuable suggestion! We have provided more details regarding the multi-agent discussion pipelines to better demonstrate the quality of our annotation results. Specifically, we uncover the details of the multi-agent discussion pipeline from five perspectives:
>
> + **Annotation Procedure:** (1) we first group the collected videos with similar sources and types (e.g. same user ID or benchmark subcategory) in a batch (we use a batchsize of 64); (2) Then we run the multi-agent discussion pipelines event-by-event to annotate each video in the batch (all prompts for the three phases, i.e. proposal, discussion, and judge are provided in Appendix.B8). We use six surrogate MLLM-based agents for annotation and provide the configuration in Table-r. 5 below.  (3) Then we ask human verifiers to sample a subset from each batch to review their explanation quality and decide whether to reject the batch and re-annotate. Specifically, grouping similar videos in a batch ensures they have similar annotation quality or shared issues, improving efficiency and reducing manual costs. If a batch has been rejected twice, then we discard this batch to exclude it from the dataset.
> + **Effectiveness:** As shown in Table-r. 4, human verifiers validate 3247 batches in total, with a low first-time rejection rate of 13.79%, demonstrating the effectiveness of our pipeline. Among the re-annotated batches, 23.7% (6784 videos) were rejected again and discarded to ensure the overall quality of the dataset.
> + **Efficiency:** The multi-agent pipeline iterates in a close-loop manner through three phases, i.e., _proposal_, _discussion_, and _judge_ to gradually reach a high-quality annotation. The results in Table-r. 4 denotes that our pipeline can efficiently produce a high-quality annotation for most unsafe videos in 1-2 iterations, while the rejected videos incur more iterations due to the videos are more ambiguous and harder for the agents to achieve consensus.
> + **Human Perspective Alignment:** We mainly guarantee the quality of the explanations through a close-loop multi-agent discussion and judge feedback, and further ask humans to select those explanations that align with their values during verification. To quantitatively verify the alignment with the human perspective, we design an experiment where we split 20 batches not used during training and prepare a pair of responses for each video where one is **the final annotation resulted from our pipeline** and the other one is directly using GPT-4o to provide the annotation. Then we adopt the following two metrics:
>     +  Implicit reward from the preference-aligned SafeWatch model, ranked by the log-likelihood ratio: $\log \frac{\pi(y_1|x)}{\pi_{\text{ref}}(y_1|x)} > \log \frac{\pi(y_2|x)}{\pi_{\text{ref}}(y_2|x)}$
>     +  Rankings provided by human reviewers
>
> The results are shown in Table-r. 6, which indicates that both the DPO model and human verifiers preferred annotations from our pipeline over GPT-4o in approximately 90% of cases, validating strong alignment with human preferences.

---

> > ### Author Response · Authors · 2024-11-22
> > **Response to Reviewer YtKs (Part 4)**
> >
> > Table-r. 4 Statistics of the multi-agent data curation process
> > |                   | #Batches                          | Avg #Iterations                          |
> > |-----------------------------|-------------------|------------------------|
> > | Total            | 3247      | 1.89      |
> > | Rejected    | 448 | 2.30      |
> > | Discarded | 106       | 2.87      |
> >
> > Table-r. 5 Configuration of the annotation surrogate model
> > |                   | #Parameters                          | #Frames                          |
> > |-----------------------------|-------------------|------------------------|
> > | Chat-univi            | 13B     | Dynamic      |
> > | VideoLLaMA2    | 7B | 8      |
> > | InternVL2-8B | 8B       | 8      |
> > | InternVL2-26B | 26B       | 8      |
> > | MiniCPM-Llama3-V-2_5  | 8B       | 1      |
> > | Cambrian-1  | 13B       | 1      |
> >
> >
> >
> >
> > Table-r. 6 Ratio of the annotation using our pipeline being chosen over direct GPT-4o annotation
> > |                   | Chosen #samples                          | Chosen ratio (%)                         |
> > |-----------------------------|--------------|-----------------------------|
> > | DPO model           | 1155 |90.23      |
> > | Human verifier   | 1093 |85.39 |
> >
> >
> > We hope that the above more detailed explanation regarding the multi-agent data curation pipeline and our dataset statistics have helped develop a more in-depth understanding of the quality of our annotation results and have addressed your concerns.
> >
> >
> > > **Q1**: Could you provide some evaluation results for Safety-aware Event Sampling? Currently, its effectiveness or limitations are unclear.
> >
> >
> > Sure, we have provided more thorough clarification and additional results regarding the training and evaluation of the Safety-aware Event Sampling model in our response to W1 above. Specifically, we demonstrate that our model is superior in providing a grounded segmentation for the unsafe events in our task.
> >
> >
> > > **Q2**:  Is SafeWatch-Bench truly a video benchmark? Specifically, does it truly require reasoning across multiple frames for a model to achieve high performance?
> >
> >
> > Yes, SafeWatch-Bench is a benchmark specifically designed to evaluate MLLMs in identifying unsafe content and providing detailed reasoning within the context of video guardrail tasks. Achieving high performance on this benchmark requires models to reason across multiple frames, particularly on the test set, where we have carefully curated challenging samples that require safety-related reasoning capabilities.
> >
> > Several case studies illustrating this are provided in Figures 16 and 17 in the appendix. For instance, in the video shown in Figure 16, which depicts a person engaged in property theft, the model needs to analyze the entire video and reason about the person’s actions across different events. This is necessary to determine that the individual is stealing someone else’s bike rather than taking their own. **A single frame would not provide sufficient context for the model to make the correct guardrail decision. This highlights the need for temporal reasoning across multiple frames to address the dataset's challenges effectively**.
> >
> > Furthermore, the dataset's complexity is also reflected in Table-r. 2, where the average number of events per video is over 7, emphasizing the intricate and dynamic nature of the videos. We are happy to provide additional examples and further explanations of the videos' quality if the reviewers are further interested.
> >
> >
> >
> > > **Q3**: It appears that humans do not directly provide annotations for SafeWatch-Bench; instead, annotations are model-generated, with human reviewers checking if re-annotation is necessary. In the caption for Figure 2, you mention that the 1K test set has high-quality annotations. Were these test set annotations created directly by humans, or were they produced via the multi-agent propose-discuss-consensus pipeline?
> >
> > Thank you for this insightful question! We have outlined the multi-agent propose-discuss-consensus pipeline and human verification pipeline in our response to W2 and W3 above. Specifically, the 1420 videos in the test set were first annotated by the multi-agent pipeline, yet each of them is further reviewed by human verifiers and corrected when necessary. This way we can ensure the absolute accuracy and high quality of the test set for the subsequent evaluations to be meaningful. To further illustrate the quality of our annotations, we have included several examples from the test set in Figures 16, 17 in the appendix.

---

> > > ### Author Response · Authors · 2024-11-22
> > > **Response to Reviewer YtKs (Part 5)**
> > >
> > > > **Q4**: Could you describe how the preference pairs were curated for the preference post-tuning stage? Additionally, how were the challenging benign examples—those easily identified by humans but likely to mislead guardrail models—selected? A detailed explanation of these curation processes would be helpful.
> > >
> > > Thank you for this question! We have detailed the data curation for the preference post-tuning stage in our response to W2 above. Specifically, we curate the rejected explanations from two sources (1) **offline collection**: the wrong, non-specific, or overly long explanations that we discarded during the multi-agent propose-discuss pipeline; (2) **online sampling:** we run our model to infer through 5K diverse videos in the training set and collect those samples with wrong answer. And we use the corresponding ground-truth explanations as the chosen response in the pair. This process results in 60K problem-centric preference pairs and we fine-tuned the model using DPO.
> > >
> > > Furthermore, for the challenging benign samples, we curate such videos for each category on platforms with stricter censorship. For example, for sexual video, we collected videos from better-monitored platforms such as TikTok and YouTube with _corresponding keywords_. While explicit content is not available on these platforms, they offer numerous _borderline videos_ that can serve as benign examples, such as videos featuring _young females dancing_ or _individuals in minimal clothing_ (but not sufficient to be considered as NSFW content). While humans can easily identify such content as benign, overly conservative guardrail models might misclassify them, leading to high false positive rates.
> > > To address this issue, we carefully curated such **challenging benign examples** for each category by selecting borderline videos from the relevant platforms and datasets. Since these platforms are monitored by humans, we can rely on their videos as benign samples, which significantly reduces our workload. This approach not only improves SafeWatch's decision boundaries but also ensures the dataset remains challenging, fostering better model performance on nuanced cases.
> > >
> > >
> > >
> > > > **Q5**: What is the quality of the synthetic videos generated by the GenAI models? Are they accurately aligned with the unsafe prompts? Was any video filtering applied to filter out misaligned videos?
> > >
> > >
> > > Thank you for this question! We totally agree that the quality of synthetic videos is indeed a concern, especially for existing datasets that use less advanced models like Stable video diffusion or HiGen. To handle this issue carefully, we rely on more advanced models such as CogVideoX which can produce videos in much higher quality and align better with the unsafe prompts. For data filtering, we leverage the data annotation pipeline to provide a description for the synthetic videos, and we leverage GPT-4o as a judge to determine if the videos have essentially covered the key points specified in the prompts and discard those videos that are unsatisfactory. This process filters out 57.3\% of the synthetic videos, and we use the rest of the high-quality videos for training and evaluation. We demonstrate 16 examples of the synthetic videos curated through this pipeline in each category of our dataset and compare them with existing datasets in Fig. 20.
> > >
> > >
> > >
> > > > **Q6**:  Could you clarify your model training data recipe? Are the unsafe videos used in stage-1 training and the guardrail tasks in stage-2 training drawn from the instruction-tuning dataset within SafeWatch-Bench? Additionally, how many samples are used for each stage, task, and dataset?
> > >
> > >
> > > We really appreciate the reviewer's interest in our training recipe. We have provided a detailed clarification of the training pipeline and dataset statistics for each of the three training stages in our response to W2 and W3 above. Specifically, in stage-1 we draw 80K safety-related samples from the SafeWatch-Bench instruction-tuning dataset and 30K normal videos from ShareGPT4Video, and after multi-task data augmentation, we obtain 220K training samples for SFT.  In stage-2, we fine-tune the model on all the 199K guardrail-related videos drawn from our instruction-tuning dataset using four types of guardrail task prompts specified in Appendix.B8. In stage-3, we curate 60K problem-centric preference pairs using the approach specified above and fine-tuned the model using DPO. For more details, please refer to our response above.
> > >
> > >
> > >
> > > > **Q7**: Which layers are tuned in the preference post-tuning stage?
> > >
> > >
> > > We fully fine-tune all the layers of the model during the preference post-tuning stage with 8 H100 GPUs. While fully fine-tuning is critical in the first two stages, for stage three, we observed that LoRA fine-tuning with rank 128 achieves similar guardrail accuracy compared to full fine-tuning. However, the explanation quality is slightly higher when using fully fine-tuning, which is why we opted for this approach for all three stages.

---

> > > > ### Author Response · Authors · 2024-11-22
> > > > **Response to Reviewer YtKs (Part 6)**
> > > >
> > > > > **Q8**: In Table 3 of Appendix A.1, there is a column named "Temporal location". What does it mean?
> > > >
> > > >
> > > > Thank you for noticing this unique feature of our dataset! The "temporal location" column specifies the exact timestamps (i.e., start and end times) for each event, as provided by the safety-aware event sampling model. This feature is a direct result of our annotation pipeline and is a distinguishing aspect of our dataset, as very few existing datasets offer this level of temporal precision. Several examples of temporal locations are illustrated in Figures 16 and 17 in the appendix.
> > > >
> > > >
> > > > > **Q9**: What is the SFT Baseline mentioned in Figure 5 and Table 5?
> > > >
> > > >
> > > > Thank you for the insightful question! The SFT baseline in Figure 5 and Table 5 refers to the InternVL2-8B model directly fine-tuned on the same dataset without incorporating novel modules such as PEPE or PAP. Specifically in Figure 5, instead of using the policy-video relevance score as the principle to measure token importance, we randomly discard the corresponding amount of video tokens for the SFT baseline. As shown in the left two panels of Figure 5, SafeWatch maintains a performance drop of less than 1% even when discarding up to 95% of video tokens, whereas such pruning significantly hurts the guardrail performance of the SFT baseline. This demonstrates the effectiveness of the policy-relevance score in providing a reliable signal for token pruning, enabling SafeWatch to retain robust performance while being more efficient with fewer tokens. In Table 5, SafeWatch also outperforms the SFT baseline with its novel components and specialized training recipe.
> > > >
> > > >
> > > > > **Q10**: It was claimed that prior datasets lack detailed descriptions of the videos, suggesting that SafeWatch-Bench offers a detailed description for each video. Is that correct?
> > > >
> > > > Yes, compared to prior datasets, SafeWatch-Bench provides significantly more detailed descriptions and guardrail explanations for each video. We discussed the dataset's quality in detail in our response to W2 above. Also as shown in Table-r. 2, the average explanation length in SafeWatch-Bench is approximately 80 words, reflecting the depth and specificity of our annotations. Furthermore, we provide additional case studies in Fig. 16, 17 in the updated appendix which compares the SafeWatch-Bench explanations with those from existing datasets. These comparisons highlight that our annotations are more specific, content-focused, and practical, whereas annotations in prior datasets are often ambiguous and less actionable.
> > > >
> > > >
> > > > > **Q11**: There is unsafe content in SafeWatch-Bench. Will SafeWatch and SafeWatch-Bench be released? If so, how do you plan to ensure their proper use?
> > > >
> > > >
> > > > Please see our response below in the ethics review.
> > > >
> > > >
> > > > > **Ethics Concerns**:The authors claim data contributions, but there is unsafe content in SafeWatch-Bench.
> > > >
> > > >
> > > > We really appreciate the reviewer for raising this concern. While our proposed guardrail model SafeWatch has been trained on a dataset containing sexually explicit, harmful, and violent content, we will not release SafeWatch-Bench in its entirety. Instead, we will release a carefully curated subset that meets the following criteria: (1) Extremely unsafe content, such as instances of child abuse, has been removed; (2) All human identities have been blurred to protect privacy; (3) Only content permitted under copyright laws is included, ensuring the data cannot be misused.
> > > > Furthermore, we will actively maintain the dataset to comply with evolving regulations and to address any samples users may find to violate their rights.
> > > >
> > > >
> > > > We hope the additional details about our dataset and training pipeline, along with the expanded explanations, have addressed your concerns. Please don’t hesitate to let us know if there’s anything else you’d like to discuss or if further clarification is needed.
> > > >
> > > > If all your concerns have been resolved, we would really appreciate it if you could consider raising the score to help us share this work with a broader community.
> > > >
> > > > Thank you again for your time and thoughtful feedback!
> > > >
> > > > Best regards,
> > > >
> > > > Submission #9949 Authors

---

> > > > > ### Comment · Reviewer_YtKs · 2024-11-27
> > > > > **Great rebuttal.**
> > > > >
> > > > > I've reviewed the revised submission and carefully read the authors' responses to all reviewers. The authors' rebuttal has adequately addressed my concerns. Thanks for the detailed responses!
> > > > >
> > > > > From my experience, the open-source VLMs utilized during the data annotation phase, such as Chat-UniVi and VideoLLaMA2, suffer significantly from hallucinations and struggle with long, complex instructions. I'm impressed by how effectively they followed the detailed instructions shown in Figures 37 and 38, resulting in such high-quality annotations, as evidenced by the authors. I strongly encourage the authors to open-source their implementation of the multi-agent consensus video annotation framework if possible.
> > > > >
> > > > > Nevertheless, the quality of the submission has significantly improved, and I am happy to increase my rating accordingly.

---

> > > > > > ### Author Response · Authors · 2024-11-27
> > > > > > **Thank you for your response!**
> > > > > >
> > > > > > We sincerely appreciate your thoughtful feedback and kind acknowledgment of the improvements! Your suggestions and questions have been invaluable in refining our work. And sure, we will ensure to open-source our implementation, including the multi-agent consensus video annotation framework and the model training and inference pipeline, to contribute to advancing the AI safety community. Thank you again for your support!

---

### Official Review · Reviewer_yJiC · 2024-11-01

**Soundness:** 1
**Presentation:** 2
**Contribution:** 2
**Rating:** 6
**Confidence:** 3

**Summary:**

SAFEWATCH is a new video moderation system that efficiently identifies harmful content and explains its decisions. It features two main innovations: PEPE (for parallel policy encoding) and PAP (for selective video analysis), both designed to make the system faster and more accurate. The researchers also developed SAFEWATCH-BENCH, a large dataset containing 2 million videos across six categories of harmful content, which they used to train and test their system.

**Strengths:**

1. The authors contribute a very large-scale benchmark for video security
2. The authors propose the PEPE algorithm, which can mitigate positional bias in the input
3. The authors propose the PAP algorithm, which maintains high recognition accuracy while reducing inference costs

**Weaknesses:**

1. The working mechanism of the PEPE algorithm lacks detailed theoretical explanation or experimental validation. The authors conduct ablation experiments to prove the effectiveness of the PEPE algorithm, but they don’t provide sufficient proof of its underlying principles. In lines 293-297, the authors claim that the PEPE algorithm can provide independent representations for each policy, which can alleviate the position bias problem in MLLM mentioned in lines 266-269. **However, regarding this claim, there are neither experimental designs nor mathematical proofs to support it. I have doubts about whether the mechanism behind the algorithm truly aligns with the authors' claims.**


2. There is a lack of explanation regarding the effectiveness of the multi-agent propose-discuss pipeline mentioned in line 105. The authors mention in lines 105-106 that they use a novel pipeline for data annotation, but there is limited discussion about this pipeline. In the pipeline-related content, the authors do not cite any references, and based on my knowledge of related work, this pipeline has not been used in any previous work, making this the first work to employ this pipeline. Given this, **I am uncertain whether this pipeline can provide sufficiently high-quality annotation results**, and the authors have not provided any quality analysis of the annotation results.

**Questions:**

1. Could the authors design corresponding experiments and proofs to demonstrate that the mechanism producing the algorithm's effects aligns with their claims that the PEPE algorithm can "allow each policy to be encoded independently and in parallel" and that "equivalent positional embedding ensures that different policies are treated without bias"?
2. Could the authors provide some quantitative analysis of the annotation quality? Can this pipeline approach human-level annotation quality? Compared to annotation by a single LLM/VLM, what advantages does incorporating Multi-agent Discussion bring?

**Details Of Ethics Concerns:**

The example in the upper left corner of Figure 3 may have 2 issues.

1. This type of data likely comes from real copyrighted videos, which may involve copyright infringement.
2. The authors did not blur or mask faces in their examples, which could raise privacy concerns.

---

> ### Author Response · Authors · 2024-11-22
> **Response to Reviewer yJiC (Part 1)**
>
> Dear Reviewer yJiC,
>
> Thank you very much for these valuable suggestions to help improve our paper! We are really grateful for your appreciation of the novelty of our work and its contribution to enhancing the trustworthiness of video generation and video security! Below we have provided a point-by-point response to your questions and hope they could help address your concerns.
>
>
> > **W1**: The working mechanism of the PEPE algorithm lacks detailed theoretical explanation or experimental validation. The authors conduct ablation experiments to prove the effectiveness of the PEPE algorithm, but they don’t provide sufficient proof of its underlying principles. In lines 293-297, the authors claim that the PEPE algorithm can provide independent representations for each policy, which can alleviate the position bias problem in MLLM mentioned in lines 266-269. However, regarding this claim, there are neither experimental designs nor mathematical proofs to support it. I have doubts about whether the mechanism behind the algorithm truly aligns with the authors' claims.
>
> We totally agree with the reviewer that a more in-depth analysis is required to validate the underlying effect of PEPE. Therefore, we have designed two additional experiments to separately prove that _PEPE allows each policy to be encoded independently and in parallel_ and _equivalent positional embedding ensures that different policies are treated without bias_. Specifically, we provide the additional evaluation results in Table-r. 1 below and in Fig.6, 7 in the updated appendix. We detail our experiment design and findings point-by-point.
>
> + **Independent, parallel policy encoding:** We design the first experiment by permuting each policy across different positions in the input and analyzing their attention score. Ideally, we would expect **the attention score to be invariant to the policy position with independent, parallel encoding** and have a constant attention score for each policy. Specifically, we randomly select a video flagged by both _Sexual_ and _Violence_ and depict the attention score curves in Fig. 6 in the appendix. The results indicate that **the policy attention scores of SafeWatch indeed preserve constant**, verifying that PEPE has eliminated the policy interdependency by decomposing the policy guidelines into several independent blocks and applying them with equivalent position embedding. We note that while the curves are not perfectly constant due to a pair of special tokens in between each policy (which is position-sensitive), which might incur some unavoidable but small interdependent patterns that can be omitted as a _structural noise_. In contrast, InternVL2-8B showed a very strong positional bias where the policies in earlier position tend to have higher attention weights in general. The curves of the model without PEPE also indicate that policies permuted among different positions may result in completely different attention scores, further indicating severe interdependencies between policies in the absence of PEPE. By independently encoding policies this way, PEPE effectively eliminates the spurious interdependency between policies and enhances the robustness of the guardrail result.
> + **Equivalent positional embedding eliminates bias:** We design another experiment by investigating the correlation of the policy attention score with both the policy position (represented by linear line vector) and the policy category (represented by one-hot vector) over the SafeWatch-Bench dataset. We evaluate the correlation with both Pearson Correlation Coefficient (PCC) and Spearman’s Rank Correlation Coefficient (SRCC), and we provide the additional evaluation results in Table-r. 1 below and Fig. 7 in the appendix. Specifically, **the policy attention score encoded with PEPE showed a very low correlation to the policy position ($\leq1\%$), and a reasonably strong correlation to the correct policy category. This means that given a video input which violates a specific policy, the attention corresponding to that policy is higher, which aligns with our expectations**. This demonstrates that PEPE effectively mitigates positional bias while improving the model's interpretability. In contrast, models without PEPE showed a strong correlation between policy attention scores and policy position, while being largely irrelevant to the correct policy category. This highlights the presence of significant positional bias in those models. Furthermore, our findings indicate that increasing the model scale does not mitigate this bias effectively.
> In summary, PEPE has proven to be an effective approach to addressing positional bias, ensuring higher interpretability by aligning attention with the correct policy category.

---

> > ### Author Response · Authors · 2024-11-22
> > **Response to Reviewer yJiC (Part 2)**
> >
> > Table-r. 1 Assessment of the correlation between the attention score of each policy chunk and the policy position and the policy category, separately. We investigate their correlations with attention scores using both Pearson Correlation Coefficient (PCC) and Spearman’s Rank Correlation Coefficient (SRCC).
> > | Model              | Policy Position (PCC ↓) | Policy Position (SRCC ↓) | Policy Category (PCC ↑) | Policy Category (SRCC ↑) |
> > |--------------------|--------------------------|---------------------------|--------------------------|---------------------------|
> > | InternVL2-8B       | -0.90                   | -0.93                     | 0.01                    | 0.00                     |
> > | InternVL2-26B      | -0.82                   | -0.86                     | 0.12                    | 0.07                     |
> > | **SafeWatch**      | **-0.094**              | **-0.076**                | **0.73**                | **0.66**                 |
> >
> > In response to the reviewer's concern, we also provide a mathematical proof to demonstrate that by independently encoding policy, PEPE can eliminate their positional bias.
> >
> > **Proof**: We model the position bias from a causal perspective where the model spuriously prioritizes certain policies based on their positional $Z$ rather than their semantic content, constituting the following causal graph:
> >
> > $$
> > Z \rightarrow T \rightarrow A \rightarrow Y
> > $$
> >
> > where $Z$ is the positional index of the policy which is the spurious factor that contributes to the bias; $T$ denotes the policy embeddings, which can be decomposed into content-dependent $T^{Z, \perp}$ and position-dependent components $T^{Z \wedge A}$ that propagates to influence the attention scores $A$, and $Y$ denotes the final guardrail output.
> > Ideally, $A$ and $Y$ should be independent of $Z$ and solely depend on the content of the policies and video. Specifically, we aim to satisfy:
> > $$
> > A \perp Z \quad \text{and} \quad Y \perp Z.
> > $$
> >
> >
> > Specifically, the attention of the $i^\text{th}$ policy and $j^\text{th}$ video token is:
> > $$
> > \mathcal{A}(i, j) = \text{softmax}\left( \frac{\hat{Q}_i \cdot \text{RoPE}(\pi^i) \hat{K}_j \cdot \text{RoPE}(v^j)^\top}{\sqrt{d}} \right),
> > $$
> >
> > where for traditional encoding $\text{RoPE}(\pi^i)$ differ for each policy based on their positional indices $Z$, which introduces dependencies between $A$ and $Z$, propagating positional bias into the model's outputs. However, PEPE applies equivalent positional embedding on all policy chunks such that $\text{RoPE}(\pi^i) = \text{RoPE}(\pi^j)$, ensuring that $Z$ does not affect how different policy attends to the video, removing spurious dependency $T^{Z \wedge A}$. Mathematically, this yields:
> > $$
> > A = f(T^{Z, \perp}), \quad A \perp Z.
> > $$
> >
> > This further ensures the guardrail outputs $Y$ to be independent of $Z$:
> > $$
> > Y = g(A), \quad Y \perp Z.
> > $$
> >
> > Specifically, the derived result $A \perp Z$ is also empirically validated by the result in Table-r. 1 where the correlation between $A$ and $Z$ is negligible ($\leq 1\%$), and that shuffling the order of input policies does not affect $Y$, demonstrating the robustness of SafeWatch to spurious positional changes.
> >
> > We hope the above two additional experiment results and the mathematical proof have helped address your concern regarding the underlying effect of PEPE. We would love to engage in a more in-depth discussion if you have further questions.

---

> > > ### Author Response · Authors · 2024-11-22
> > > **Response to Reviewer yJiC (Part 3)**
> > >
> > > > **W2**:  There is a lack of explanation regarding the effectiveness of the multi-agent propose-discuss pipeline mentioned in line 105.
> > >
> > >
> > > Thank you for this valuable observation and suggestion! We have provided more details regarding the multi-agent discussion pipelines to better demonstrate the quality of our annotation results. Specifically, we analyze the effectiveness of the multi-agent discussion pipeline from the following five perspectives:
> > >
> > > + **Annotation Procedure:** (1) we first group the collected videos with similar sources and types (e.g. same user ID or benchmark subcategory) in a batch (we use a batchsize of 64); (2) Then we run the multi-agent discussion pipelines event-by-event to annotate each video in the batch (all prompts provided in Appendix.B8); (3) Then we ask human verifiers to sample a subset from each batch to review their explanation quality and decide whether to reject the batch and re-annotate. Specifically, grouping similar videos in a batch ensures they have similar annotation quality or shared issues, improving efficiency and reducing manual costs. If a batch has been rejected twice, then we discard this batch to exclude it from the dataset.
> > > + **Effectiveness:** As shown in Table-r. 2, human verifiers validate 3247 batches in total, with a low first-time rejection rate of 13.79%, demonstrating the effectiveness of our pipeline. Among the re-annotated batches, 23.7% (6784 videos) were rejected again and discarded to ensure the overall quality of the dataset.
> > > + **Efficiency:** The multi-agent pipeline iterates in a close-loop manner through three phases, i.e., _proposal_, _discussion_, and _judge_ to gradually reach a high-quality annotation. The results in Table-r. 2 denote that our pipeline can efficiently produce a high-quality annotation for most unsafe videos in 1-2 iterations, while the rejected videos incur more iterations due to the videos are more ambiguous and harder for the agents to achieve consensus.
> > > + **Human Perspective Alignment:** We mainly guarantee the quality of the explanations through a close-loop multi-agent discussion and judge feedback, and further ask humans to select those explanations that align with their values during verification. To quantitatively verify the alignment with the human perspective, we design a toy experiment where we split 20 batches not used during training and prepare a pair of responses for each video where one is **the final annotation resulted from our pipeline** and the other one is directly using GPT-4o to provide the annotation. Then we adopt the following two metrics:
> > >     +  Implicit reward from the preference-aligned SafeWatch model, ranked by the log-likelihood ratio: $\log \frac{\pi(y_1|x)}{\pi_{\text{ref}}(y_1|x)} > \log \frac{\pi(y_2|x)}{\pi_{\text{ref}}(y_2|x)}$
> > >     +  Rankings provided by human reviewers
> > >
> > > The results are shown in Table-r. 3, which indicates that both the DPO model and human verifiers preferred annotations from our pipeline over GPT-4o in approximately 90% of cases, validating strong alignment with human preferences.
> > > + **Case Study:** We further provide several qualitative analyses of the annotation pipeline in Fig. 15 in the appendix, where we demonstrate how our pipeline can effectively correct errors through close-loop discussion and feedback and yield annotation with human-like quality. We further demonstrate several cases by comparing our annotation with previous datasets in Fig. 16, 17 in the appendix, where we can clearly denote that our annotation is much more clarified and in-depth than the previous benchmarks. We also demonstrate some generative video samples in SafeWatch-Bench in Fig. 20 which are more diverse and have higher-quality than those in existing benchmarks.
> > >
> > >
> > >
> > > Table-r. 2 Statistics of the multi-agent data curation process
> > > |                   | #Batches                          | Avg #Iterations                          |
> > > |-----------------------------|-------------------|------------------------|
> > > | Total            | 3247      | 1.89      |
> > > | Rejected    | 448 | 2.30      |
> > > | Discarded | 106       | 2.87      |
> > >
> > >
> > > Table-r. 3 Ratio of the annotation using our pipeline being chosen over direct GPT-4o annotation
> > > |                   | Chosen #samples                          | Chosen ratio (%)                         |
> > > |-----------------------------|--------------|-----------------------------|
> > > | DPO model           | 1155 |90.23      |
> > > | Human verifier   | 1093 |85.39 |
> > >
> > >
> > > We hope that the above more detailed explanation regarding the multi-agent data curation pipeline and our dataset statistics have helped develop a more in-depth understanding of the quality of our annotation results and have addressed your concerns.

---

> ### Author Response · Authors · 2024-11-22
> **Response to Reviewer yJiC (Part 4)**
>
> > **Q1**: Could the authors design corresponding experiments and proofs to demonstrate that the mechanism producing the algorithm's effects aligns with their claims that the PEPE algorithm can "allow each policy to be encoded independently and in parallel" and that "equivalent positional embedding ensures that different policies are treated without bias"?
>
> Thank you for this thoughtful question! In response, we have designed two additional experiments to separately verify that _PEPE allows each policy to be encoded independently and in parallel_ and _equivalent positional embedding ensures that different policies are treated without bias_. We provide the additional evaluation results in Table-r. 1 above and also in Fig.6, 7 in the updated appendix. We also provide a theoretical analysis from a causal learning perspective to prove the effect of PEPE in our response to W1. For more details, please refer to our response to W1.
>
>
> > **Q2**: Could the authors provide some quantitative analysis of the annotation quality? Can this pipeline approach human-level annotation quality? Compared to annotation by a single LLM/VLM, what advantages does incorporating Multi-agent Discussion bring?
>
>
> We really appreciate the reviewer's concern about the annotation quality of our pipeline, and we further uncover the effectiveness of the multi-agent discussion pipeline from five perspectives in our response to W2. Specifically, we provide detailed explanations regarding the _annotation procedure_, _effectiveness_, _efficiency_, _human perspective alignment_, and several _case study_ to explain how this approaches human-level annotation quality. Essentially, we ask human verifiers to validate the quality of a subset of each batch of the annotation and the results in Table-r. 2 indicate over 86\% annotated batches were accepted by human verifiers. We further design a specific experiment where we use an automatic model and ask human verifiers to compare the quality of the annotations resulting from our pipeline and GPT-4o, and we find that the automatic metric and human prefer our annotation 90.23\% and 85.39\% of the time.
>
> Specifically, we conclude the reason that multi-agent discussion can achieve human-level annotation quality to be three-fold:
> + **Comprehensiveness:** Multi-agent discussion integrates the capabilities of different LLMs/VLMs, where they can compensate with each other to provide a more comprehensive and in-depth understanding of the video. Furthermore, our discussion mechanism encourages agents to more logically and thoroughly reason about the contents in the video in order to convince other agents.
> + **Consistency:** By encouraging agents to reach consistency through discussion, we can effectively search for a commonly agreed intersection of each agent's opinion, ensuring the accuracy of the final annotation.
> + **Close-loop feedback:** By integrating a more powerful judge model to review the dicussion and provide feedback, we can gradually correct errors and improve the accuracy and quality of the annotation through iterative discussions. In contrast, the quality of single LLM/VLM annotations cannot be guaranteed as they lack this iterative refinement and error correction mechanism.
>
>
> > **Ethics Concerns**: The example in the upper left corner of Figure 3 may have 2 issues. (1) This type of data likely comes from real copyrighted videos, which may involve copyright infringement; (2) The authors did not blur or mask faces in their examples, which could raise privacy concerns.
>
> We thank the reviewer for raising these concerns. In response, (1) We will release a subset of the dataset that will not contain copyrighted videos. As for concerns related to the release of the models, since this model is released for exclusive research purposes and implies a transformative usage of the video data (our model uses the video to produce improved textual descriptions, and cannot reproduce the video itself since it is not a video generative model) this usage of copyrighted data falls under the fair use doctrine. For further compliance, we will impose licensing that will prevent misuse of the released models. (2) We will ensure that all human identities, including faces, are blurred or masked in both the examples and the released dataset to mitigate any potential privacy issues.
>
> We hope the additional details about our dataset and training pipeline, along with the expanded explanations, have addressed your concerns. Please don’t hesitate to let us know if there’s anything else you’d like to discuss or if further clarification is needed.
>
> If all your concerns have been resolved, we would greatly appreciate it if you could consider raising the evaluation score to help us share this work with the broader community.
>
> Thank you again for your time and thoughtful feedback!
>
> Best regards,
>
> Submission #9949 Authors

---

> > ### Author Response · Authors · 2024-11-26
> > **We would like to hear back from reviewer yJiC**
> >
> > Dear reviewer yJiC,
> >
> > We sincerely thank you for your thoughtful feedback, which has been invaluable in improving our work! Based on your suggestion to provide more clarification on the PEPE mechanism and the data curation pipeline, we have carefully improved our paper by designing three additional experiments, conducting a detailed theoretical analysis, and providing more thorough justifications.
> >
> > As other reviewers have also acknowledged, our work addresses critical challenges posed by video generative models through a novel video guardrail model with well-designed modules and a comprehensive safety benchmark. And your feedback has been instrumental in enhancing the clarity of our presentation and the potential impact of our work.
> >
> > Please let us know if you have further questions or suggestions and we would like to discuss with you and further improve our work to build the safe AI community together! Thank you again!!

---

> > > ### Comment · Reviewer_yJiC · 2024-11-26
> > >
> > > Thank you for your detailed response. After reading your detailed response, I have decided to increase your score.

---

> > > > ### Author Response · Authors · 2024-11-27
> > > > **Thank you for your response!**
> > > >
> > > > Thank you for your positive feedback and for updating your scores! We sincerely appreciate the time and effort you’ve invested in helping us refine and improve our work!!

---

### Official Review · Reviewer_QsGZ · 2024-11-02

**Soundness:** 3
**Presentation:** 4
**Contribution:** 3
**Rating:** 8
**Confidence:** 3

**Summary:**

This paper presents an MLLM-based video guardrail model that takes into account safety policies to provide a multi-label video content output including explanation, considering both the safety policies and the video content. The proposed model comprises two plug-and-play modules to improve latency of the guard rail model and mitigate positional biases by breaking down the safety guidelines. This work also introduces a benchmark for video guardrailing using multi-agent consensus and comparison across existing MLLMs.

**Strengths:**

- The paper addresses a critical topic by proposing guardrails for video MLLMs based on defined safety policies, which is timely and important with the rise of MLLMs.
- It introduces a baseline model built upon the InternVL2-8B backbone and leverages two plug-in modules to (1) improve latency during training and inference, and (2) reduce positional biases related to the policy order.
- The benchmark provides a comprehensive evaluation of existing MLLMs on video guardrailing tasks, demonstrating the model’s effectiveness across six safety policy categories, covering 30 subtopics.

**Weaknesses:**

- Details about the training and testing splits within the benchmark are insufficient, leaving questions about data partitioning.
- The authors should clarify if any videos were discarded during dataset curation due to multi-agent discussion pipelines not reaching a consensus or human verification disagreements on final explanations. This clarification could shed light on the multi-agent approach's effectiveness in generating explanations that align with human perspectives, especially given video content's subjective nature.

**Questions:**

- SFT Baseline: Could the authors provide additional context for the "SFT baseline" mentioned in Figure 5?
- Inference Cost: What accounts for the increase in inference cost with additional few-shot examples, as illustrated in Figure 5?

**Details Of Ethics Concerns:**

The proposed model for video guardrailing, along with the benchmark for evaluating safety policies in existing MLLMs, addresses sensitive content types. Given the nature of the videos and policies (safety categories) evaluated, it is essential to examine the work for potential biases, privacy concerns, potential harms, and legal compliance.

---

> ### Author Response · Authors · 2024-11-22
> **Response to Reviewer QsGZ (Part 1)**
>
> Dear Reviewer QsGZ,
>
> Thank you very much for your valuable suggestions to help improve our paper! We deeply appreciate your recognition of the novelty of our work and its critical contribution to handling the risks brought by the rise of MLLMs. Below, we have provided a detailed, point-by-point response to your questions, and hope these could help address your concerns.
>
> > **W1:** Details about the training and testing splits within the benchmark are insufficient, leaving questions about data partitioning.
>
> We totally agree with the reviewer that the details on training/testing splits are critical for understanding the mechanism and effectiveness of our work. In response, we have provided additional details on the dataset partitioning and statistics in Table-r. 1 below, as well as Fig. 10-20 in the updated appendix. Specifically,
>
> + **Dataset partition:** For more effective training and evaluation, we use the 200K videos verified by humans via batch sampling and select 1420 videos to consist the testing set for benchmarking (830 real-world videos, 590 generative videos), and use the rest of 199604 videos for training. Specifically, we aim to ensure the diversity of the videos and a balanced coverage of all categories in the test set (more detailed distribution is shown in Fig. 10, data sample case study in Fig. 16, 17, 20 in the appendix).
> + **Data usage in each training stage:** We detail the data usage for each of the three fine-tuning stages, and we demonstrate the corresponding model's performance in each stage in Table-r. 2:
>     + **Multi-task Guardrail Training:** In this stage, we randomly sample 80K guardrail-related videos we collected and 30K normal videos in ShareGPT4Video annotated by GPT-4v, and then we augment the original annotations into multiple tasks including captioning, VQA, and video guardrails, resulting in 220K training samples. This stage aims to train the model to develop general guardrail capabilities while preserving a broad understanding of general video content, effectively mitigating catastrophic forgetting and overfitting to guardrail-specific videos.
>     + **Adaptive-Pruning Training:** In this stage, we solely fine-tune the model on all the 199K guardrail-related videos using four types of guardrail task prompts specified in Appendix.B8. This stage aims to train the model to extract essential information from a subset of more informative video tokens via PAP and downstream the model for specialized guardrail tasks.
>     + **Preference Post-tuning:** In this stage, we aim to further improve the quality of explanations. Specifically, we curate the rejected explanations from two sources (1) **offline collection**: the non-specific or overly long explanations that we discarded during the multi-agent propose-discuss pipeline; (2) **online sampling:** we run the model from the previous stage to infer through 5K diverse videos in the training set and collect those samples with wrong answer. And we use the corresponding ground-truth explanations as the chosen pair. This process results in 60K problem-centric preference pairs and we fine-tuned the model using DPO.
>
> + **Additional Dataset Statistics:** We provide more statistics to better understand our dataset and training procedure. Specifically,
>      + The **average video length** in the training and testing set is 57.49 and 61.12 secs, respectively (more detailed distribution in Fig. 11 in the appendix), with the longest video up to 90 minutes, ensuring a comprehensive coverage of unsafe videos with different scales (e.g. short videos, movies).
>      + The **average explanation length** in the training and testing set is 80.2 and 73.16 words, ensuring detailed and in-depth reasoning of the guardrail result (more detailed distribution in Fig. 12, two case studies in Fig. 16, 17 in the appendix).
>      + The **average number of events** generated by our safe-aware event sampler model in the training and testing set is 7.33 and 7.4. This reflects the high complexity and challenging nature of the videos in the guardrail task.
>      + The **average ratio of multi-labeled videos** (i.e., videos flagged with multiple guardrail categories) is 24.7% in the training set and 28.57% in the testing set, further demonstrating the dataset's diversity and challenging nature.

---

> > ### Author Response · Authors · 2024-11-22
> > **Response to Reviewer QsGZ (Part 2)**
> >
> > Table-r. 1 Additional statistics of the SafeWatch-Bench Dataset. We provide details for both training and testing configurations, including total video samples, average video length (seconds), explanation length (word count), number of events, and the ratio of multi-labeled videos.
> > | Datset Configuration                  | **Training**                              | **Testing**                               |
> > |-----------------------------|-------------------------------------------|-------------------------------------------|
> > | Total Videos            | 199604       | 1420        |
> > | Average Video Length    | 57.49 | 61.12  |
> > | Average Explanation Length | 80.20       | 73.16         |
> > | Average Number of Events      | 7.33           | 7.40           |
> > | Average Ratio of Multi-label        | 24.70          | 28.57           |
> >
> >
> > Table-r. 2 Performance of SafeWatch during each training stage
> > | Model              | Accuracy | Explanation | Adaptability |
> > |--------------------|--------------------------|---------------------------|--------------------------|
> > | InternVL2-8B       | 29.1                   | 5.25                     | 26.8                    |
> > | Stage-1      | 63.9              | 5.84               | 69.7                |
> > | Stage-2      | 67.3              | 6.12                | 74.9                |
> > | Stage-3      | **72.6**              | **7.17**                | **78.0**                |
> >
> > > **W2:** The authors should clarify if any videos were discarded during dataset curation due to multi-agent discussion pipelines not reaching a consensus or human verification disagreements on final explanations. This clarification could shed light on the multi-agent approach's effectiveness in generating explanations that align with human perspectives, especially given video content's subjective nature.
> >
> > Thank you for this insightful question! We completely agree that more details regarding the multi-agent discussion pipelines will facilitate the understanding of the quality of our dataset and its alignment with human perspectives. We will first provide a detailed explanation of the annotation procedure. Specifically, (1) we first group the collected videos with similar sources and types (e.g. same user ID or benchmark subcategory) in a batch (we use a batchsize of 64); (2) Then we run the multi-agent discussion pipelines event-by-event to annotate each video in the batch (all prompts provided in Appendix.B8); (3) Then we ask human verifiers to sample a subset from each batch to review their explanation quality and decide whether to reject the batch and re-annotate. Specifically, grouping similar videos in a batch ensures they have similar annotation quality or shared issues, further improving efficiency and reducing manual costs. If a batch has been rejected twice, then we discard this batch to exclude it from the dataset. Specifically, we analyze the effectiveness of the multi-agent discussion pipeline from three-folds:
> > + **Effectiveness:** As shown in Table-r. 3, human verifiers validate 3247 batches in total, with a low first-time rejection rate of 13.79%, demonstrating the effectiveness of our pipeline. Among the re-annotated batches, 23.7% (6784 videos) were rejected again and discarded to ensure the overall quality of the dataset.
> > + **Efficiency:** The multi-agent pipeline iterates in a close-loop manner through three phases, i.e., _proposal_, _discussion_, and _judge_ to gradually reach a high-quality annotation. The results in Table-r. 3 denotes that our pipeline can efficiently produce a high-quality annotation for most unsafe videos in 1-2 iterations, while the rejected videos incur more iterations due to the videos being more ambiguous and harder for the agents to achieve consensus.
> > + **Human Perspective Alignment:** We mainly guarantee the quality of the explanations through (1) our close-loop multi-agent discussion and judge feedback, and (2) further ask humans to verify and select those explanations that align with their values. To quantitatively verify the alignment with the human perspective, we design an experiment where we split 20 batches not used during training and prepare a pair of responses for each video where one is **the final annotation resulted from our pipeline** and the other one is directly using GPT-4o to provide the annotation. Then we adopt the following two metrics:
> >     +  Implicit reward from the preference-aligned SafeWatch model, ranked by the log-likelihood ratio: $\log \frac{\pi(y_1|x)}{\pi_{\text{ref}}(y_1|x)} > \log \frac{\pi(y_2|x)}{\pi_{\text{ref}}(y_2|x)}$
> >     +  Rankings provided by human reviewers
> >
> > The results are shown in Table-r. 4, which indicates that both the DPO model and human verifiers preferred annotations from our pipeline over GPT-4o in approximately 90% of cases, validating strong alignment with human preferences.

---

> > > ### Author Response · Authors · 2024-11-22
> > > **Response to Reviewer QsGZ (Part 3)**
> > >
> > > Table-r. 3 Statistics of the multi-agent data curation process
> > > |                   | #Batches                          | Avg #Iterations                          |
> > > |-----------------------------|-------------------|------------------------|
> > > | Total            | 3247      | 1.89      |
> > > | Rejected    | 448 | 2.30      |
> > > | Discarded | 106       | 2.87      |
> > >
> > >
> > > Table-r. 4 Ratio of the annotation using our pipeline being chosen over direct GPT-4o annotation
> > > |                   | Chosen #samples                          | Chosen ratio (%)                         |
> > > |-----------------------------|--------------|-----------------------------|
> > > | DPO model           | 1155 |90.23      |
> > > | Human verifier   | 1093 |85.39 |
> > >
> > >
> > > While video content inherently involves subjectivity, we hope these clarifications demonstrate the robustness and high quality of our multi-agent pipeline annotations. We appreciate your suggestion and are happy to address any further questions.
> > >
> > > > **Q1:** SFT Baseline: Could the authors provide additional context for the "SFT baseline" mentioned in Figure 5?
> > >
> > > Thank you for the insightful question! The SFT baseline in Figure 5 refers to the InternVL2-8B model directly fine-tuned on the same dataset without incorporating novel modules such as PEPE or PAP. Specifically, instead of using the policy-video relevance score as the principle for measuring token importance, we randomly discard the amount of video tokens for the SFT baseline. As shown in the left two panels of Figure 5, SafeWatch maintains a performance drop of less than 1% even when discarding up to 95% of video tokens, whereas such pruning significantly hurts the guardrail performance of the SFT baseline. This demonstrates the effectiveness of the policy-relevance score in providing a reliable signal for token pruning, enabling SafeWatch to retain robust performance while being more efficient with fewer tokens.
> > >
> > > > **Q2:** Inference Cost: What accounts for the increase in inference cost with additional few-shot examples, as illustrated in Figure 5?
> > >
> > > The increase in inference cost primarily results from the **higher number of input tokens**, as each few-shot example adds approximately 380 tokens. This leads to a longer prefilling phase, which accounts for the rise in overall inference time.
> > >
> > > While all methods improve with additional few-shot examples, SafeWatch demonstrates a slightly steeper performance gain compared to both GPT-4o and the SFT baseline. This suggests that SafeWatch has better scalability with respect to few-shot examples and stronger transferability for guardrail tasks.

---

> > > > ### Author Response · Authors · 2024-11-22
> > > > **Response to Reviewer QsGZ (Part 4)**
> > > >
> > > > > **Ethics Concerns:** The proposed model for video guardrailing, along with the benchmark for evaluating safety policies in existing MLLMs, addresses sensitive content types. Given the nature of the videos and policies (safety categories) evaluated, it is essential to examine the work for potential biases, privacy concerns, potential harms, and legal compliance.
> > > >
> > > > We really appreciate the reviewer for raising this concern!
> > > >
> > > > As for potential biases and harms, a portion of the contents of our dataset are already publicly available online and often used to train the models we aim to guardrail. Besides, we explicitly consider the bias concern when collecting the videos, where we first collect user IDs that are proportionate to the user demographical statistics across the social media platforms to ensure a balanced coverage of the unsafe content being distributed publicly. Such a mechanism can promote a balanced and debiased representation across different demographic groups, which is also crucial for training SafeWatch to be effectively and practically deployed (a demographic distribution of our dataset is provided in Fig. 14 in the appendix). And since we deliberately collect this type of data to mitigate the impact this type of malicious content may have on existing models, our usage of this data is ethical and aims at fostering a safer and more responsible deployment of AIs. The release of our data will also be restricted to individuals with a public research profile and affiliated to reputable institutions in order to reduce the chances of their misuse.
> > > > As for legal compliance, we will exclusively release data that is not copyright protected and that does not infringe local laws. When applying to access the data, practitioners will have to indicate their provenance and intended use. Based on their provenance, we will remove some categories of the dataset that may be deemed to be illegal to distribute in specific countries (e.g. sexually explicit content).
> > > > As for privacy concerns, we will pre-process our data using a PII detector in order to remove any possible private data, and we will ensure to blur all the human identities such as faces before we release the data to the users. We will also remain available to remove any data samples that users may deem to violate their privacy.
> > > >
> > > > We hope the additional details about our dataset and training pipeline, along with the expanded explanations, have addressed your concerns. Please don’t hesitate to let us know if there’s anything else you’d like to discuss or if further clarification is needed.
> > > >
> > > > If all your concerns have been resolved, we would sincerely appreciate it if you could consider raising the score to help us share this work with a broader community!
> > > >
> > > > Thank you again for your time and thoughtful feedback!
> > > >
> > > > Best regards,
> > > >
> > > > Submission #9949 Authors

---

> > > > > ### Comment · Reviewer_QsGZ · 2024-11-24
> > > > > **Thank you**
> > > > >
> > > > > I thank the authors for a comprehensive response for the concerns raised, which have been addressed. I will prefer to retain my original score.

---

> > > > > > ### Author Response · Authors · 2024-11-24
> > > > > > **Thank you for your response!**
> > > > > >
> > > > > > Thank you for your prompt and thoughtful feedback!! We are delighted to hear that the concerns raised have been resolved! As there is still time remaining in the discussion period, we would be happy to address any additional concerns or questions the reviewer might have to further enhance the quality of the paper. If all concerns have indeed been resolved, we would sincerely appreciate it if the reviewer could kindly consider further raising the score to help us share this work with a broader audience! Thank you!

---

> > > > > > > ### Comment · Reviewer_QsGZ · 2024-11-25
> > > > > > >
> > > > > > > Having read the other reviews and responses, I am willing to increase my score given the comprehensive rebuttal response by the authors.

---

> > > > > > > > ### Author Response · Authors · 2024-11-25
> > > > > > > > **Response to Reviewer QsGZ**
> > > > > > > >
> > > > > > > > We sincerely appreciate your thoughtful feedback, which has been invaluable in refining our paper. We are also grateful for your kind acknowledgment of the improvements and your efforts to help us share this work with a wider audience. Thank you very much!

---

### Author Response · Authors · 2024-11-22
**A Summary of Rebuttal and Paper Revisions**

We sincerely thank all the reviewers and chairs for their valuable time and thoughtful feedback on our work. We are glad that all the reviewers agree on its technical novelty and critical contribution to addressing the risks given by the rise of multimodal foundation models. We have carefully addressed each reviewer’s concerns and questions point-by-point and made corresponding updates in the revisions. Below is a summary of the additional clarifications and evaluation results included in our rebuttal:

- Following reviewer yJiC's suggestions, we have designed two additional experiments and conducted theoretical analyses to validate the principles of PEPE. The results, provided in Table 10 and Figures 6–7 in the appendix, verify that the working mechanism of PEPE indeed aligns with our expectations in the paper.

- Following reviewer QsGZ, yJiC, and YtKs's suggestions, we have designed additional experiments and provided various case studies to demonstrate the effectiveness and efficiency of our proposed multi-agent data annotation pipeline in achieving human-like annotation quality. The results are shown in Table 18-19 and Figures 15–20 in the appendix.

- Following reviewer YtKs's suggestion, we have designed an additional evaluation of the safe-aware event sampling to demonstrate its effectiveness in providing accurate unsafe event grounding for the guardrail task. The results are shown in Table 9 in the appendix.

- To address reviewer QsGZ, YtKs, and PM58's concerns, we have provided a more comprehensive explanation of the dataset statistics and the training pipeline, including data details, prompts, configurations, and intermediate results for each training stage. These are presented in Table 9, Table 11-20, and Figures 10–14 in the appendix and elaborated on in our individual responses. All the prompts used in this work are provided in the Appendix. B8.

- To address all reviewers' ethical concerns, we have outlined an actionable plan including releasing only a subset of the videos that are not copyrighted, with all the extremely harmful categories such as child abuse removed, and with all the human identities blurred to ensure that potential biases, privacy concerns, harms, and legal compliance are thoroughly and responsibly addressed.

Thank you again for your thoughtful suggestions and efforts in helping us improve SafeWatch. We hope our response adequately addresses your concerns, and we look forward to engaging in further discussions during the follow-up period!

Best regards,

Submission #9949 Authors

---

### Meta-Review · Area_Chair_wZiE · 2024-12-20

**Metareview:**

This work introduces a new video moderation method that efficiently identifies harmful content and explains its decisions. The work has two major innovations: PEPE (for parallel policy encoding) and PAP (for selective video analysis), and both are well designed to make the system fast and accurate. Reviewers pointed out that the design choices are well-founded, which follow best practices for efficient MLLM construction. More importantly, this work addresses an important problem of study, with meaningful contributions. Thus, all reviewers recommend acceptance. AC agrees with the reviewers.

**Additional Comments On Reviewer Discussion:**

The authors provided a comprehensive response for the concerns raised by reviewers (including additional explanations and experiments). Reviewers are generally satisfied with the reply.

---

### Decision · Program_Chairs · 2025-01-22

Accept (Poster)